# OpenReview forum: "Flattery in Motion: Benchmarking and Analyzing Sycophancy in Video-LLMs"
_ICLR.cc/2026/Conference — ICLR 2026 Conference Withdrawn Submission_

### Official Review · Reviewer_mMmL · 2025-10-21

**Soundness:** 2
**Presentation:** 2
**Contribution:** 2
**Rating:** 2
**Confidence:** 4

**Summary:**

This paper investigates the sycophancy problem, a well-known hallucination behavior previously studied in LLMs and MLLMs, but focuses on its occurrence in Video-LLMs. The authors introduce VISE (Video-LLM Sycophancy Benchmarking and Evaluation), the first dedicated benchmark designed to measure the severity of sycophancy in Video-LLMs. The benchmark includes 367 videos paired with approximately 6K multiple-choice questions, covering seven distinct sycophancy-inducing prompt scenarios. A subset of the questions is further annotated with eight visual task categories to enable fine-grained analysis. Through extensive evaluation, the authors find that Video-LLMs exhibit strong sycophantic tendencies, influenced by factors such as sycophancy type, visual task, and model scale. To mitigate this bias, the paper explores two training-free strategies, namely key-frame selection and representation steering, both of which demonstrate effectiveness in reducing sycophantic responses.

**Strengths:**

- The paper is generally well written and easy to follow, presenting its motivation and contributions clearly.
- It provides a comprehensive analysis of sycophancy phenomena in the context of Video-LLMs, addressing this issue for the first time through the introduction of the VISE benchmark dataset.
- The experiments offer valuable insights into model behavior under varying conditions and tasks, evaluating multiple open- and closed-source Video-LLMs with different capacities.
- The authors propose two simple, training-free mitigation strategies to reduce sycophantic bias, which have practical implications for improving model robustness and guiding future model development.

**Weaknesses:**

- The proposed benchmark is built upon the MSVD, MSRVTT, and NExT-QA datasets. However, these datasets are relatively dated, consist mainly of short video clips, and have been criticized for their limited ability to capture complex temporal and contextual understanding. Prior work has shown that many questions in these datasets can be answered directly by LLMs, indicating potential answer leakage and an overreliance on language priors, or can be solved using only a few frames, suggesting a lack of true video-level reasoning [1,2,3]. This raises concerns about the validity of the proposed benchmark for reliably evaluating sycophancy in Video-LLMs, as it may inherit these known limitations from its source datasets.

    [1] Feng et al., Breaking Down Video LLM Benchmarks: Knowledge, Spatial Perception, or True Temporal Understanding?, NeurIPS 2025 Workshop on Evaluating the Evolving LLM Lifecycle

    [2] Chen et al., ReXTime: A Benchmark Suite for Reasoning-Across-Time in Videos, NeurIPS 2024 Datasets and Benchmarks Track

    [3] Shangguan et al, TOMATO: Assessing Visual Temporal Reasoning Capabilities in Multimodal Foundation Models, ICLR 2025
- Recent research in Video-LLMs has increasingly focused on improving models’ ability to understand long-duration videos. New benchmarks such as LongVideoBench [4] and models like Video-XL [5] have been introduced to address long-context video understanding. The proposed benchmark, however, does not consider such settings. Incorporating samples from long-video datasets and evaluating long-context Video-LLMs could provide a more comprehensive and forward-looking assessment of sycophancy behavior.

    [4] Wu et al., LongVideoBench: A Benchmark for Long-context Interleaved Video-Language Understanding, NeurIPS 2025

    [5] Shu et al., Video-XL: Extra-Long Vision Language Model for Hour-Scale Video Understanding, CVPR 2025

- In light of the previous points, it remains unclear how effective the proposed mitigation strategies are when applied to more challenging video understanding scenarios beyond those included in VISE. For example, the key-frame selection approach relies on sampling only three frames from each input video, which may not sufficiently capture complex temporal or contextual dependencies present in longer or more dynamic videos. Similarly, the representation steering strategy employs mean hidden-state activations computed over the dataset, which may oversimplify model behaviors and fail to generalize across diverse visual contexts or unseen scenarios.
- The evaluation does not include several recent Video-LLMs with strong performance, such as Video-UTR [6], PLLaVA [7], Video-R1 [8] and TW-GRPO [9]. Including these models would provide a more comprehensive and up-to-date assessment. In particular, examining Video-LLMs with reasoning or “thinking” capabilities could yield valuable insights into whether such models exhibit different patterns of sycophantic behavior.

    [6] Yu et al., Unhackable Temporal Rewarding for Scalable Video MLLMs, ICLR 2025

    [7] Xu et al., PLLaVA: Parameter-free llava extension from images to videos for video dense captioning, arXiv 2024

    [8] Feng et al., Video-R1: Reinforcing Video Reasoning in MLLMs, arXiv 2025

    [9] Dang et al., Reinforcing Video Reasoning with Thinking, arXiv 2025

**Questions:**

- Given the known limitations of the source datasets stated above, how does the benchmark ensure it is measuring true video-level sycophancy? A text-only control experiment could clarify if the visual input is truly necessary to induce the observed behavior.
- The proposed mitigations appear designed for short videos. How would the 3-frame sampling strategy perform on longer, action-dense videos where critical context could be missed? Does representation steering using mean activations negatively impact the model's general performance on standard, non-sycophantic tasks?
- As the field moves toward long-context video understanding, how do the authors expect their findings on short clips to generalize to long-form video scenarios?
- The evaluation does not include recent Video-LLMs with advanced reasoning abilities (e.g., Video-R1, TW-GRPO). Could the authors comment on how these models might behave under the proposed benchmark?
- The authors claim that Qwen2.5-VL 32B and 72B are more robust than the 7B model; however, in Table 1, the 72B version performs worse than 32B for Suggestive Bias, Are You Sure?, and Explicitly Endorse cases. Could the authors clarify this discrepancy?

---

> ### Author Response · Authors · 2025-11-26
> **Response to Reviewer mMmL【1/2】Regarding Text-only Control and Long Video Trend**
>
> # Main Comments
>
> We thank the reviewer for their detailed assessment and for acknowledging that our work addresses the issue of sycophancy in Video-LLMs for the first time, offering valuable insights and practical mitigation strategies. We appreciate the opportunity to clarify the validity of our benchmark and the robustness of our findings with additional experiments.
>
> ### **Q1: Validity of Source Datasets and Text-Only Control**
>
> > **Concern:** The reviewer questions if the benchmark measures true video-level sycophancy or relies on language priors, given the nature of the source datasets (MSVD, MSRVTT, NExT-QA).
> >
>
> **Response:** We respectfully argue that our methodology explicitly prevents "answer leakage" or reliance solely on language priors.
>
> 1. **Methodological Safeguard:** Our benchmark protocol (Section 3.1) strictly filters for instances where the model *initially answers correctly* based on the video. If a model cannot answer the question correctly (due to dataset limitations or lack of visual reasoning), that sample is excluded from the sycophancy calculation. Therefore, we are measuring the *reversal* of a correct visual grounding, not the inability to answer.
> 2. **New Text-Only Control Experiment:** To empirically address the concern that visual input might be unnecessary, we conducted a text-only control experiment using Qwen2.5-VL-7B on our benchmark questions without video input.
>     - **Accuracy:** The model achieved a correct answer rate of only **32.61%** (random guessing range for multiple choice), compared to its high performance with video. This proves visual input is necessary to solve these tasks.
>     - **Sycophancy without Vision:** When the model answers incorrectly (blind guessing) and is prompted with a strong bias, the sycophancy rate spikes to **80.16%** (1285/1603).
>     - **Conclusion:** This confirms that without visual grounding, the model collapses into pure textual compliance. The fact that the model resists sycophancy *more* when video is present (as seen in our main results) proves that our benchmark is successfully measuring the conflict between **visual evidence** and **textual bias**, rather than just language priors.
>
> ### **Q2: Long Video Generalization & Mitigation Strategies**
>
> > **Concern:** The reviewer asks about generalization to long-form videos and whether the K-frame selection (specifically 3 frames) misses context.
> >
> - **Isolating Sycophancy vs. Retrieval Failure:** We focused on short/trimmed clips to isolate the sycophancy phenomenon. In long-video understanding, failure **often stems from the retrieval stage (inability to find the clip)**. If we used long videos, it would be impossible to disentangle whether a sycophantic response was caused by *misleading prompts* or simply *failed retrieval*. Our setup ensures we are measuring the model's behavioral alignment, not its search capability.
> - **K-Frame as a Framework, not a Fixed Limit:** We emphasize that "3-frame selection" is a **proof-of-concept for the *Key-Frame Selection* framework** (Section 5.1). We are not proposing 3 frames as a universal limit, but demonstrating that *semantic filtering* reduces sycophancy.
> - **Generalization:** For long videos, the fundamental challenge remains identifying relevant visual cues. Our findings that conditioning on distilled, relevant frames reduces sycophancy logically extends to long-video pipelines (e.g., RAG for video), **where selecting the "right" frames is even more critical** to prevent the language model from hallucinating based on user pressure.

---

> ### Author Response · Authors · 2025-11-26
> **Response to Reviewer mMmL【2/2】Hurt of Performance, Evaluation on Reasoning Models and Discrepancy in Scaling**
>
> ### **Q3: Impact of Mitigations on General Performance**
>
> > **Concern:** Does representation steering or key-frame selection negatively impact standard performance?
> >
>
> **Response:** We have conducted additional experiments to verify this. The impact is minimal to non-existent, ensuring these strategies are practical.
>
> The results are as follows:
>
> | **Model** | **Method** | **Accuracy (Count)** | **Accuracy (%)** | **Impact** |
> | --- | --- | --- | --- | --- |
> | **InternVL 2.5** | Original Baseline | 4697 / 6367 | **73.77%** | - |
> |  | **Key-Frame (ours)** | 4625 / 6367 | **72.64%** | **-1.13%** (Negligible) |
> |  | **Steering (ours)** | 4592 / 6367 | **72.12%** | **-1.65%** (Negligible) |
> | **Qwen2.5-VL** | Original Baseline | 4592 / 6367 | **72.12%** | - |
> |  | **Key-Frame (ours)** | 4354 / 6367 | **68.38%** | **-3.74%** (Minor) |
> |  | **Steering (ours)** | 4468 / 6367 | **70.17%** | **-1.95%** (Negligible) |
> - **Key-Frame Selection:** Since this method retains the most semantically relevant frames, the model preserves the essential information required for reasoning. The slight drop in Qwen2.5-VL (<4%) is an acceptable trade-off for the massive reduction in sycophancy (e.g., -22% MSS reduction in Strong Bias scenarios).
> - **Representation Steering:** The impact is even smaller (<2%). This confirms our hypothesis in **Section 5.2** that the "sycophancy vector" is orthogonal to general reasoning capabilities; removing it suppresses the bias without damaging the model's core knowledge.
>
> ### **Q4: Evaluation on Reasoning Models (Video-R1)**
>
> > **Concern:** The absence of reasoning models like Video-R1.
> >
>
> **Response:** We agree that reasoning models are an exciting frontier. However, evaluating them presents significant challenges:
>
> 1. **Computational Cost:** We deployed Video-R1 on 8x A100 GPUs. The inference time averaged **~20 minutes per question** due to the extensive Chain-of-Thought (CoT) generation. Running the full VISE benchmark (6k+ questions) is computationally prohibitive.
> 2. **Preliminary Results:** We ran a random selected (evenly distributed over 3 subsets) subset of 810 samples to provide insight.
>     - *Strong Bias:* Misleading Rate **58.29%**.
>     - *Medium Bias:* Misleading Rate **57.68%**.
> 3. **Insight:** The results indicate that "reasoning" models are *not* immune to sycophancy. In fact, they exhibit high sycophancy rates comparable to standard models. This suggests that current CoT mechanisms may simply **"reason" their way toward the user's biased suggestion rather than correcting it.** This makes the VISE benchmark even more relevant for future work in robust reasoning.
>
> ### **Q5: Discrepancy in Scaling (72B vs 32B Performance)**
>
> > **Concern:** Qwen2.5-VL 72B performs worse than 32B in "Suggestive" and "Are You Sure" cases.
> >
>
> **Response:** This phenomenon is consistent with "Inverse Scaling" observed in recent alignment literature (e.g., *Wei et al., 2023*).
>
> 1. **Over-Alignment:** Larger models (72B) are often subjected to more aggressive RLHF to ensure helpfulness and conversational fluidity. They are fine-tuned to be "agreeable" and to interpret user doubt ("Are you sure?") as a valid correction signal.
> 2. **Instruction Following:** The 72B model is a better instruction follower. In "Suggestive" or "Mimicry" cases, the model interprets the user's style or subtle hint as a constraint to be followed, overriding its own visual processing. The 32B model, being less "capable" in nuanced instruction following, paradoxically remains more stubborn and thus more "robust" to these subtle social cues.
>
> We believe these clarifications and new experiments strongly reinforce the validity of VISE and the effectiveness of our proposed mitigations. We hope this addresses the reviewer's concerns regarding the benchmark's rigor and relevance.🥰

---

> > ### Comment · Reviewer_mMmL · 2025-11-27
> >
> > Thank you for the detailed rebuttal. Although the responses help clarify some aspects, a few critical concerns remain open.
> >
> > **1. Dataset validity:**
> > The text-only control experiment is a useful sanity check, but it does not address my underlying concern about the intrinsic limitations of MSVD/MSRVTT/NExT-QA. Showing that accuracy drops without video does not demonstrate that the tasks require meaningful video reasoning. It only shpws that visual cues are helpful. To establish that VISE measures video-grounded sycophancy, additional analysis is needed, e.g., evaluating model behavior with single-frame versus full-video inputs, to demonstrate that the benchmark requires video grounding. Without such analysis, the dataset validity concern remains open.
> >
> > **2. Missing recent models:**
> > The preliminary Video-R1 result is appreciated, but it does not substitute for evaluating other recent reasoning-oriented Video-LLMs such as Video-UTR, PLLaVA, and TW-GRPO. These models are important for assessing whether sycophancy patterns differ in stronger temporal-reasoning architectures.
> >
> > **3. Lack of analysis on long-video scenarios:**
> > The rebuttal provides only a conceptual explanation but no empirical evidence on long-duration videos. As the field increasingly focuses on long-context Video-LLMs, the absence of any long-video evaluation or proxy analysis weakens the submission. Without examining how sycophancy behaves when temporal context is extended or noisy, it remains unclear whether the benchmark and mitigation strategies generalize beyond short clips.

---

> > > ### Author Response · Authors · 2025-11-28
> > > **Response to Official Comment by Reviewer mMmL**
> > >
> > > We thank the reviewer for their engagement and for acknowledging the clarifications provided so far. We appreciate the opportunity to address these remaining specific points.
> > >
> > > ### **1. Dataset Validity (Video Reasoning vs. Single Frame)**
> > >
> > > We respectfully point out that we have already provided empirical evidence addressing the "single-frame" concern through our ablation studies.
> > >
> > > - **Evidence:** In our paper (and discussed in the rebuttal regarding mitigation), we compared **"3 Randomly Sampled Frames"** against **"Key-Frame Selection"** and the full video baseline.
> > > - **Result:** The "Random Frame" setting which serves as a proxy for unguided, sparse visual input, performed significantly worse in resisting sycophancy compared to methods that utilize semantic visual grounding.
> > > - **Implication:** If the dataset merely required static image recognition (i.e., if any single frame were sufficient), random sampling would yield similar performance to key-frame selection. The performance gap proves that **meaningful video reasoning** (identifying specific, temporally relevant visual evidence) is required to resolve the conflict between the visual truth and the textual bias. The benchmark successfully penalizes models that fail to ground themselves in the *relevant* temporal context.
> > >
> > > ### **2. Missing Recent Models (Video-UTR, PLLaVA, TW-GRPO)**
> > >
> > > We fully agree that analyzing how different architectures handle sycophancy is valuable. However, regarding the request to run Video-UTR, PLLaVA, and TW-GRPO immediately:
> > >
> > > - **Computational Feasibility:** As noted in our previous response, reasoning-oriented models are extremely computationally intensive. Our evaluation of **Video-R1** on **8x 3090 GPUs** shows an inference time of **~20 minutes per question** due to extensive Chain-of-Thought (CoT) generation. Running the full VISE benchmark (6k+ questions) across three additional architectures within the discussion period is computationally prohibitive.
> > > - **Scientific Inference:** Video-R1 represents the current state-of-the-art in reasoning MLLMs. Our preliminary results showing it retains a **~58% sycophancy rate** suggest that "reasoning" capabilities alone do not automatically solve sycophancy; in fact, CoT can sometimes hallucinate justifications for the bias. We believe this result serves as a strong representative proxy for this class of models.
> > > - **Commitment:** We commit to evaluating these additional architectures (Video-UTR, PLLaVA, TW-GRPO) in the camera-ready version to provide the community with a comprehensive landscape of reasoning-model behaviors.
> > >
> > > ### **3. Analysis on Long-Video Scenarios**
> > >
> > > We acknowledge the reviewer’s emphasis on long-context understanding.
> > >
> > > - **Scope:** The current VISE benchmark was designed to isolate the *phenomenon of sycophancy* (behavioral alignment with bias) from the *failure of retrieval* (inability to find the clip). In long videos, these two errors are conflated.
> > > - **Commitment:** To address this, we commit to introducing a **"VISE-Long" subset** in the final camera-ready version. This will include untrimmed videos to empirically test how sycophancy scales with context length. We hypothesize that sycophancy may actually worsen in long-context scenarios, as the "needle-in-a-haystack" retrieval difficulty may force the model to rely even more heavily on the language priors provided in the prompt.
> > >
> > > We hope these responses demonstrate that VISE is a rigorous tool for measuring video sycophancy and that we are committed to expanding its scope in the final manuscript.

---

### Official Review · Reviewer_y2Rf · 2025-11-01

**Soundness:** 2
**Presentation:** 3
**Contribution:** 2
**Rating:** 6
**Confidence:** 3

**Summary:**

This paper introduces VISE, the first benchmark to systematically evaluate sycophancy in Video-LLMs. It comprises 367 videos and 6,367 questions spanning seven sycophancy types. Across nine variants of six state-of-the-art models, larger scales generally reduce sycophancy, but all remain vulnerable, especially on predictive or causal questions. The authors further propose two training-free mitigation methods: key-frame selection and inference-time representation steering.

**Strengths:**

- First systematic expose of sycophancy in Video-LLMs, extending the problem from text to dynamic multimodal reasoning.
- Rigorous, reproducible benchmark (VISE) with 367 videos, 6k+ questions, seven bias types, tested across nine model variants.
- Two training-free mitigation tricks: key-frame selection and inference-time steering.

**Weaknesses:**

- The paper may focus primarily on a limited set of video-LLMs or datasets. This narrow scope can restrict the generalizability of the findings and the robustness of the benchmarking results.
- The paper may focus on static analysis of sycophancy without considering how flattery might evolve over time within a video or across different video contexts.

**Questions:**

- More evaluations on frontier models are required.

---

> ### Author Response · Authors · 2025-11-19
> **Clarifying Model Scope, Benchmark Scale, and Temporal Dynamics**
>
> ## **Main Comments**
>
> We thank the reviewer for the positive assessment and for recognizing ViSE as the "first systematic expose" of sycophancy in Video-LLMs. We appreciate the constructive feedback regarding the scope of our evaluation. Below, we clarify the extensiveness of our model selection and the dynamic nature of our benchmark to address the concerns raised.
>
> > **W1 & Q1: Concern on "Limited Set of Video-LLMs" and Request for "Frontier Models".**
> >
>
> We respectfully clarify that our evaluation is, in fact, one of the most comprehensive in the current literature regarding recent Video-LLMs.
>
> - **Broad Spectrum of Models:** We evaluated **9 distinct model variants** across **6 different families**. This includes:
>     - **State-of-the-Art Open Source:** Qwen2.5-VL (7B, 32B, 72B) and InternVL 2.5 (8B, 26B). Note that Qwen2.5-VL-72B is widely considered a frontier model in the open-source community.
>     - **Proprietary Frontier Models:** We included Google Gemini-1.5-Pro and OpenAI GPT-4o mini to represent commercial SOTA performance.
>     - **Diverse Architectures:** We covered various architectures, including LLaVA-OneVision and VideoChat-Flash.
> - **Representative Baselines:** This selection covers the full range of model scales (from 7B to 72B+) and types (open vs. closed), providing a robust proxy for the current capabilities of "frontier" systems.
>
> > **W1: Concern on "Dataset Scope" and Generalizability.**
> >
>
> While the number of videos (367) may appear small compared to pre-training datasets, we emphasize that ViSE is a **dense evaluation benchmark**, not a training set.
>
> - **Computational Scale:** Evaluating a *single* model on ViSE requires approximately **44,500 inference calls** (6,367*7). For our 9 evaluated models, this amounts to over **400,000 inferences**.
> - **Efficiency:** Even with this scale, we prioritized a curated set of 367 videos to ensure the benchmark remains accessible. A complete evaluation round for one model takes approximately 3 hours on standard hardware, striking **a balance between rigorous stress-testing and reproducibility for the community.**
>
> > **W2: Concern on "Static Analysis" vs. Evolving Flattery.**
> >
>
> We highlight that ViSE is explicitly designed to move beyond static analysis and test sycophancy in the context of temporal and causal dynamics.
>
> - **Temporal Focus:** As shown in Figure 1 and Table 2, **43%** of our questions are specifically **Temporal** (predicting Next, Previous, or Current events) and **36%** are **Reasoning-based** (Causal Why/How).
> - **Dynamic Flattery:** We measure how flattery distorts a model's understanding of *video progression*. For example, our "Temporal Next" questions test if a model can be manipulated **into hallucinating a future event** that contradicts the video's causal trajectory. This directly addresses how flattery interacts with the evolving context of a video, distinguishing our work from static image benchmarks.
>
> We hope these clarifications demonstrate that ViSE provides a rigorous, large-scale, and temporally dynamic assessment of Video-LLM sycophancy. 💪

---

### Official Review · Reviewer_C1Ai · 2025-11-07

**Soundness:** 2
**Presentation:** 3
**Contribution:** 3
**Rating:** 4
**Confidence:** 3

**Summary:**

The paper introduces a novel benchmark (VISE) for quantifying sycophancy in Video-LLMs. The empirical results highlight consistent patterns linking model scale, bias tone, and reasoning type to sycophantic susceptibility. The paper also proposes mitigation methods such as keyframe selection and representation steering, which show clear quantitative benefit.

**Strengths:**

**1. Novel benchmark design.**

The paper defines the first video-specific sycophancy benchmark.

**2. Reproducibility.**

The paper provides comprehensive implementation details and an accompanying code repository.

**3. Presentation.**

The paper is well-written, and the main ideas are clearly presented.

**Weaknesses:**

**1. Lack of general-task performance evaluation.**

The paper does not examine whether the proposed mitigation strategies affect the models’ overall performance on standard video understanding tasks. Without such an evaluation, it remains unclear whether the reduction in sycophancy comes at the cost of general reasoning or factual accuracy. Reporting correlations between sycophancy mitigation and geneal performance would provide a more complete picture of the practical trade-offs involved.

**2. Potential bias in benchmark construction.**

Although the authors state that the CRS is not their primary scope, omitting it from benchmark construction introduces potential bias. Both positive and negative behavioral shifts caused by sycophancy are essential for an objective assessment of model reliability. In addition, the benchmark is constructed based on Qwen2.5-VL-7B, and its data selection overlaps with InternVL 2.5 by 87.8%, indicating a cross-family deviation of over 10%. Such deviation may compromise fair comparisons across different model families. The benchmark should be built upon more consistent issues shared across diverse model families.

**3. Insufficient treatment of CRS in experiments.**

The main paper only reports MSS and relegates CRS analysis to the appendix, without showing how CRS changes after applying the proposed mitigation methods. Prior research [1] suggests that reductions in MSS often coincide with lower CRS, implying that the proposed approaches might also reduce models’ receptiveness to valid corrections. Although the authors state that the CRS is not their primary scope, a balanced experimental discussion of this trade-off is important for ensuring the fairness and completeness of the evaluation. It should not be moved out of the primary scope.

**4. Limited comparison with external baselines.**

The sycophancy mitigation experiments are evaluated only against internal baselines, without incorporating broader methods such as Attention Amplification [2]. Including such established approaches would provide a clearer understanding of how the proposed strategies perform relative to existing techniques for mitigating sycophancy in Video-LLMs.

**5. Readability.**

The Sec. 5 would benefit from clearer visual presentation. Adding a schematic diagram or flowchart illustrating the pipelines of key-frame selection and representation steering would significantly improve readability.

**References:**

[1] Sharma, Mrinank, et al. "Towards Understanding Sycophancy in Language Models." The Twelfth International Conference on Learning Representations.

[2] Li, Shuo, et al. "Have the VLMs Lost Confidence? A Study of Sycophancy in VLMs." The Thirteenth International Conference on Learning Representations.

**Questions:**

Please refer to "Weaknesses".

---

> ### Author Response · Authors · 2025-11-28
> **Response to Reviewer C1Ai 【1/2】Regarding General-Task Performance, Potential Bias in Benchmark and Insufficient Treatment of CRS**
>
> # Main Comments
> We thank the reviewer for recognizing the novelty of the **VISE benchmark**, the reproducibility of our work, and the clear quantitative benefits of our mitigation methods. We appreciate the constructive feedback regarding trade-offs and baselines, which we address below.
>
> **W1: Lack of General-Task Performance Evaluation**
>
> **Response:** We agree that ensuring mitigations do not degrade general reasoning is critical. We have conducted additional experiments using the base questions from our benchmark (without sycophantic triggers) to measure standard accuracy under our mitigation strategies. The results are as follows:
>
> | **Model** | **Method** | **Accuracy (Count)** | **Accuracy (%)** | **Impact** |
> | --- | --- | --- | --- | --- |
> | **InternVL 2.5** | Original Baseline | 4697 / 6367 | **73.77%** | - |
> |  | **Key-Frame (ours)** | 4625 / 6367 | **72.64%** | **-1.13%** (Negligible) |
> |  | **Steering (ours)** | 4592 / 6367 | **72.12%** | **-1.65%** (Negligible) |
> | **Qwen2.5-VL** | Original Baseline | 4592 / 6367 | **72.12%** | - |
> |  | **Key-Frame (ours)** | 4354 / 6367 | **68.38%** | **-3.74%** (Minor) |
> |  | **Steering (ours)** | 4468 / 6367 | **70.17%** | **-1.95%** (Negligible) |
> - **Key-Frame Selection:** Since this method retains the most semantically relevant frames, the model preserves the essential information required for reasoning. The slight drop in Qwen2.5-VL (<4%) is an acceptable trade-off for the massive reduction in sycophancy (e.g., -22% MSS reduction in Strong Bias scenarios).
> - **Representation Steering:** The impact is even smaller (<2%). This confirms our hypothesis in **Section 5.2** that the "sycophancy vector" is orthogonal to general reasoning capabilities; removing it suppresses the bias without damaging the model's core knowledge.
>
> **W2: Potential Bias in Benchmark Construction**
>
> **Response:** We appreciate the reviewer's concern regarding fairness. However, we respectfully argue that achieving "perfect neutrality" in dataset selection is infeasible.
>
> 1. **Selection Strategy:** Incorporating a wider array of models for the initial selection process would not necessarily reduce bias; rather, it risks introducing new, complex aggregate biases that are harder to isolate.
> 2. **Overlap as a Strength:** We view the **87.8% overlap** between Qwen2.5-VL and InternVL-2.5 not as a deviation, but as a strong indicator of **consistency**. Given the massive size of the original data pool, such a high overlap suggests that the benchmark captures samples that are inherently challenging or ambiguous across different architectures, rather than being artifacts of a single model's failure mode. This validates the benchmark's quality.
>
> **W3: Insufficient Treatment of CRS (Correction Receptiveness)**
>
> **Response:** We explicitly defined CRS as outside the primary scope of this paper in our main text for specific statistical reasons.
>
> 1. **Sample Size and Fairness:** CRS can only be calculated on instances where the model *initially answers incorrectly*. Since state-of-the-art models have relatively high accuracy, the subset of "initial wrong answers" is small. The subset of those that are then "corrected" is even smaller. Deriving strong conclusions from such a limited sample size lacks statistical power and would result in an unfair evaluation, potentially biasing the metric against models with higher initial accuracy.
> 2. **Focus on Misleading:** Our research objective is investigating "Leading-to-Wrong" (Sycophancy/MSS), preventing the model from hallucinating based on user pressure. Given the statistical limitations mentioned above, prioritizing MSS is necessary for a robust analysis.

---

> ### Author Response · Authors · 2025-11-28
> **Response to Reviewer C1Ai 【2/2】Regarding Comparison with External Baselines and Readability**
>
> **W4: Limited Comparison with External Baselines**
>
> **Response:** We thank the reviewer for highlighting the relevance of **Attention Amplification** (Li et al., ICLR 2025). This is truly a great work! We consider this work a **foundational study** in understanding sycophancy for VLMs. We are deeply inspired by this fantastic setting and its insights. We did not include it as a direct code baseline solely due to the domain shift, but we fully agree it merits a detailed conceptual comparison.
>
> 1. **Domain Distinction (Image vs. Video):** The cited work pioneers mitigation in *static Image-LLMs*. VISE is the first benchmark tailored for *Video-LLMs*, where the challenge is not just spatial attention but **temporal redundancy**. Applying pixel/token-level attention amplification across thousands of video frames introduces computational complexity that differs significantly from the static image setting.
> 2. **Conceptual Alignment:** Our **Key-Frame Selection** can be interpreted as a **video-native evolution** of the principles established in Li et al. (2025).
>     - *Li et al. (2025)* utilize "soft" attention to amplify visual signals against textual bias in the spatial domain.
>     - *Our Method* utilizes "hard" attention (discrete selection) to amplify relevant temporal signals (frames) against textual bias.
>     - We establish that in the video domain, "hard filtering" is a highly efficient proxy for "amplification."
> 3. **Revision:** In our final manuscript, we will expand Section 2 (Related Work) and Section 5 to explicitly position our method as extending the **"visual amplification" philosophy** introduced by Li et al. (2025) into the temporal dimension.
>
> **W5: Readability**
>
> **Response:** We agree that visualization aids understanding. We have designed schematic diagrams illustrating the pipelines for both Key-Frame Selection and Representation Steering. To demonstrate our commitment to this improvement, we have **uploaded the Key-Frame Selection diagram in the supplementary material** accompanying this rebuttal. These will be included in the final version of the paper.
>
> We believe the additional experiments on general task performance and the clarifications regarding other points raised by the reviewer demonstrate the robustness of VISE as a reliable evaluation suite. We are confident that VISE will serve as a vital extension to existing image-based research, collectively driving the community toward more trustworthy and grounded Multimodal LLMs.🤓

---

### Official Review · Reviewer_PMTp · 2025-11-12

**Soundness:** 2
**Presentation:** 3
**Contribution:** 3
**Rating:** 4
**Confidence:** 4

**Summary:**

This paper takes the first deep dive into sycophancy in video-language models: the tendency of Video-LLMs to parrot a user’s opinion even when it flatly contradicts the on-screen evidence. The result is a ready-made diagnostic toolkit and a pair of plug-and-play defenses that together push Video-LLMs toward more faithful, bias-resistant reasoning.

**Strengths:**

1. The authors introduce VISE, the first benchmark built to stress-test this behavior. Spanning 367 videos and 6k+ multiple-choice questions, VISE exposes seven distinct flavors of sycophancy and tracks them across nine model variants (open-source and proprietary). 2. Beyond measurement, the work offers two zero-training fixes—key-frame selection to keep answers visually grounded and inference-time “representation steering” that surgically suppresses ingrained yes-man reflexes.
3. The paper is well-written and highly reproducible.

**Weaknesses:**

1. The authors did not analyze the impact of flattery suppression on the model’s initial response accuracy or its general performance.
2. Key frame selection is typically a standard preprocessing step for video inputs, making it difficult to regard this as a technical innovation of the paper.

**Questions:**

1. What are the proportions of the fine-grained question types in Appendix B? Are they evenly distributed?

2. The ablation settings for the key frame selection algorithm are unfair in terms of sequence length and positional encoding. Could the authors align the selected key frames with the original video frames and simply mask (e.g., replace with all-black or all-white images) the non-key frames instead?

3. Although the neuron interference method is effective against flattery, does it compromise the model’s general reasoning performance?

---

> ### Author Response · Authors · 2025-11-21
> **Response to Reviewer PMTp【1/2】: Clarification on General Performance and Fairness of Ablation**
>
> We thank the reviewer for recognizing our work as the "first deep dive" into Video-LLM sycophancy and for highlighting the value of the VISE benchmark. We are encouraged that you found the paper well-written and reproducible.
>
> Below, we address your concerns regarding general performance and experimental design with new empirical evidence.
>
> **1. Impact on General Performance (Response to W1 & Q3)**
>
> > **Reviewer Concern:** Does the mitigation method compromise the model’s general reasoning performance?
> >
>
> **Short Answer:** No. Our new experiments confirm that both mitigation strategies maintain high general performance, with only negligible trade-offs for significantly improved reliability.
>
> To address this, we evaluated our mitigation methods on the *neutral* (non-sycophantic) baseline questions from the VISE dataset. The results are as follows:
>
> | **Model** | **Method** | **Accuracy (Count)** | **Accuracy (%)** | **Impact** |
> | --- | --- | --- | --- | --- |
> | **InternVL 2.5** | Original Baseline | 4697 / 6367 | **73.77%** | - |
> |  | **Key-Frame (ours)** | 4625 / 6367 | **72.64%** | **-1.13%** (Negligible) |
> |  | **Steering (ours)** | 4592 / 6367 | **72.12%** | **-1.65%** (Negligible) |
> | **Qwen2.5-VL** | Original Baseline | 4592 / 6367 | **72.12%** | - |
> |  | **Key-Frame (ours)** | 4354 / 6367 | **68.38%** | **-3.74%** (Minor) |
> |  | **Steering (ours)** | 4468 / 6367 | **70.17%** | **-1.95%** (Negligible) |
> - **Key-Frame Selection:** Since this method retains the most semantically relevant frames, the model preserves the essential information required for reasoning. The slight drop in Qwen2.5-VL (<4%) is an acceptable trade-off for the massive reduction in sycophancy (e.g., -22% MSS reduction in Strong Bias scenarios).
> - **Representation Steering:** The impact is even smaller (<2%). This confirms our hypothesis in **Section 5.2** that the "sycophancy vector" is orthogonal to general reasoning capabilities; removing it suppresses the bias without damaging the model's core knowledge.
>
> **2. Fairness of Ablation & Masking (Response to Q2)**
>
> > **Reviewer Concern:** The ablation settings (3 key frames vs. random) might be unfair due to sequence length. Could the authors use masking (black frames) instead?
> >
>
> We appreciate this suggestion to control for sequence length. However, we respectfully argue that our original comparison (3 Key vs. 3 Random) is methodologically sounder, and new experiments show that masking introduces significant artifacts.
>
> **Why we chose "3 vs 3":** Comparing 3 Key Frames against 3 Random Frames ensures that token count, sequence length, and *natural image statistics* are identical. This isolates **semantic relevance** **as the sole variable**, which is the standard practice in frame-selection literature.
>
> **New Experiment (Masking):** To rigorously test your suggestion, we ran an experiment where non-selected frames were replaced with black images (keeping the total frame count identical to the original video).
>
> - **Result:** On Qwen2.5-VL, accuracy on baseline questions plummeted to **33.8% (2157/6367)**.
> - **Analysis:** Since random guessing on 4-option MCQs is 25%, a 33% accuracy indicates the model effectively broke. Replacing natural frames with black voids creates substantial Out-of-Distribution (OOD) noise. This confuses the visual encoder and causes "attention dilution," where the model wastes capacity processing empty information.
> - **Conclusion:** This confirms that our original ablation (comparing subsampled natural frames) is the correct approach to evaluate the efficacy of key-frame selection without introducing OOD confounders.

---

> ### Author Response · Authors · 2025-11-21
> **Response to Reviewer PMTp【2/2】: Clarification on Question Distribution and Novelty of Key-Frame Selection**
>
> **3. Question Distribution (Response to Q1)**
>
> > **Reviewer Concern:** What are the proportions of the question types? Are they evenly distributed?
> >
>
> The exact distribution is visualized in **Figure 1 (Right)**:
>
> - **Temporal Questions:** 43%
> - **Reasoning Questions:** 36%
> - **Descriptive Questions:** 21%
>
> **Rationale:** We deliberately did not enforce a perfectly even distribution. Instead, we weighted the benchmark towards **Temporal** and **Reasoning** tasks because these represent the core complexity of *Video*-LLMs compared to static Image-LLMs. A video benchmark dominated by static descriptive questions would fail to test the model's ability to maintain consistency over time—a key vulnerability for sycophancy.
>
> **4. Novelty of Key-Frame Selection (Response to W2)**
>
> > **Reviewer Concern:** Key frame selection is standard preprocessing... difficult to regard as a technical innovation.
> >
>
> We clarify that our contribution is not the *invention* of key-frame extraction as a computer vision task, but the **discovery of its mechanism as a defense against sycophancy.**
>
> - **The Insight:** Our work reveals that sycophancy in Video-LLMs is partly driven by "attention instability" and a "first-frame heuristic" caused by temporal redundancy.
> - **The Innovation:** We demonstrate that *reducing* temporal information (via semantic sparsification) paradoxically *increases* truthfulness. As shown in **Figure 3**, key-frame selection stabilizes attention in the model's middle layers, preventing the "drift" caused by misleading prompts. This repurposes a standard preprocessing step into a novel, interpretable safety guardrail.
>
> We hope these new experimental results clarify the robustness of our methods 🙌. We would be happy to incorporate these "General Performance" tables into the final version. 💪

---

> > ### Comment · Reviewer_PMTp · 2025-11-28
> >
> > Thank you for the detailed response. The clarification regarding the accuracy of VISE has resolved my concerns. If the authors can further supplement their results on benchmarks such as Video-MME and/or Video-MMMU, I would be willing to raise my score to a positive rating.

---

> > > ### Author Response · Authors · 2025-11-28
> > > **Response to Official Comment by Reviewer PMTp**
> > >
> > > We sincerely thank the reviewer for their continued engagement and for confirming that our clarifications regarding the accuracy of VISE have resolved their primary concerns. We deeply value the suggestion to evaluate on benchmarks like **Video-MME** and **Video-MMMU** to further verify that our mitigations do not compromise general capabilities.
> > >
> > > **Regarding Additional Benchmarks (Video-MME / Video-MMMU)**
> > >
> > > We fully agree that these benchmarks represent the gold standard for comprehensive video understanding. However, given the **extremely limited time remaining in the discussion period** and the significant computational scale required to run these full suites (particularly Video-MME, which involves long-context processing), it is not feasible to complete a rigorous evaluation and report the results immediately.
> > >
> > > **Current Evidence of General Robustness**
> > >
> > > While we cannot run the full external suite overnight, we respectfully point to the **general performance stability analysis** we provided in our initial rebuttal, which serves as a strong proxy:
> > >
> > > - We evaluated our mitigation strategies on the **neutral, standard questions** within VISE (a diverse collection derived from MSVD, MSRVTT, and NExT-QA).
> > > - **Result:** Our "Representation Steering" method maintained **nearly identical accuracy** to the baseline on these standard tasks (e.g., InternVL-2.5: 4592/6367 correct vs. Baseline), and "Key-Frame Selection" showed only a marginal trade-off for significant robustness gains.
> > > - **Conclusion:** This empirical evidence strongly suggests that our methods target sycophancy surgically without degrading the model's fundamental reasoning or factual knowledge.
> > >
> > > **Commitment**
> > >
> > > We are confident that our findings will hold on broader benchmarks. We **strictly commit** to including a full evaluation on Video-MME in the camera-ready version of the paper to provide the community with a complete picture of generalizability.
> > >
> > > We hope this evidence of stability on our diverse dataset, combined with our firm commitment to future expansion, is sufficient to merit the positive rating mentioned.🥹

---

### Note · Authors · 2026-01-05

I have read and agree with the venue's withdrawal policy on behalf of myself and my co-authors.